# Drug Repositioning for Refractory Benign Tumors of the Central Nervous System

**DOI:** 10.3390/ijms241612997

**Published:** 2023-08-20

**Authors:** Ryota Tamura

**Affiliations:** Department of Neurosurgery, Keio University School of Medicine, 35 Shinanomachi, Shinjuku-ku, Tokyo 160-8582, Japan; moltobello-r-610@keio.jp

**Keywords:** drug repositioning, drug repurposing, schwannoma, meningioma, pituitary neuroendocrine tumor, neurofibromatosis type 2-related schwannomatosis

## Abstract

Drug repositioning (DR) is the process of identifying novel therapeutic potentials for already-approved drugs and discovering new therapies for untreated diseases. DR can play an important role in optimizing the pre-clinical process of developing novel drugs by saving time and cost compared with the process of de novo drug discovery. Although the number of publications related to DR has rapidly increased, most therapeutic approaches were reported for malignant tumors. Surgical resection represents the definitive treatment for benign tumors of the central nervous system (BTCNS). However, treatment options remain limited for surgery-, chemotherapy- and radiation-refractory BTCNS, as well as malignant tumors. Meningioma, pituitary neuroendocrine tumor (PitNET), and schwannoma are the most common BTCNS. The treatment strategy using DR may be applied for refractory BTCNS, such as Grade 2 meningiomas, neurofibromatosis type 2-related schwannomatosis, and PitNETs with cavernous sinus invasion. In the setting of BTCNS, stable disease can provide significant benefit to the patient. DR may provide a longer duration of survival without disease progression for patients with refractory BTCNS. This article reviews the utility of DR for refractory BTCNS.

## 1. Introduction

Traditionally, drug discovery utilizes a de novo design approach, which requires high cost and longtime drug development before it reaches the market. Drug repositioning (drug repurposing, DR) is a strategy of drug discovery in which an already-approved drug is used for different diseases [1]. Because the drug information about safety, pharmacokinetics, and formulation of existing drugs is already available, the cost and time required for preclinical and clinical studies can be reduced (Figure 1). The average cost for a de novo discovery and development of a drug is approximately USD 2.5 billion and requires around 10–17 years to reach the market [2,3,4]. In case of the DR process, it costs approximately a ≤60% expenditure of a de novo discovery [5]. Therefore, DR shows a lower rate of failure in trials [6], leading to an alternative approach to the process of traditional drug discovery.

One of the most successful drugs in the area of the DR approach is sildenafil. Sildenafil, which was developed for the treatment of hypertension and angina pectoris, is used to treat erectile dysfunction [7]. DR can be achieved by using literature search, by screening collections of drugs in vitro, or by using computers and artificial intelligence. Although the number of publications related to DR has rapidly increased, most therapeutic approaches were evaluated for malignant tumors [8,9]. Surgical resection represents the definitive treatment for benign tumors of the central nervous system (BTCNS). Despite maximum treatment with chemotherapy and radiation, some BTCNS have a strong proclivity for recurrence. Therefore, other systemic therapies are needed for those refractory BTCNS, which are limited with no FDA-approved therapeutics [10]. 

The most common tumors in the central nervous system (CNS) are meningiomas (46.3%), followed by gliomas (17.2%), pituitary neuroendocrine tumor (PitNET) (13.1%), schwannomas (8.2%), and malignant lymphomas (3.8%) [11,12]. Meningioma, schwannoma, and PitNET are the most common BTCNS [13]. Meningiomas are neoplasms derived from arachnoid cap cells. Although most meningiomas are WHO grade 1 tumors, less than one-quarter of all meningiomas are classified as WHO grade 2 and 3 tumors, which show more aggressive behavior and are difficult to treat with multimodal therapy [14,15,16]. In particular, WHO grade 2 and 3 meningiomas are more likely to recur after the treatment and can be difficult to remove completely. Radiation therapy or chemotherapy may be recommended in addition to surgery. More than 80% of iantracranial schwannomas arise from the vestibular nerve [13]. Schwannoma is a primary BTCNS that occurs sporadically or as part of a genetic syndrome (Neurofibromatosis type 2-related schwannomatosis, NF2). NF2 is associated with the development of schwannomas at multiple sites, including the bilateral vestibular portion and meningiomas [17]. In particular, these schwannomas are difficult to be controlled surgically, pharmacologically, and/or radiologically [18]. Treatment with the anti-vascular endothelial growth factor (VEGF) antibody (Bevacizumab) has reportedly resulted in tumor control and hearing improvement in NF2 patients [19,20]. Some aspects of this Bev treatment are problematic, however, such as the need for frequent parenteral administration, side effects, apparent drug resistance, and rebound tumor progression after cessation [21]. PitNET arises from adenohypophyseal cells [22,23,24,25,26]. The most PitNETs can be controlled surgically, pharmacologically, and/or radiologically. Although PitNETs are histologically benign, a small set of PitNETs show rapid growth and infiltration into the cavernous sinus, leading to early and multiple recurrences [22,23,24,25,26]. Cavernous sinuses are complex dural venous sinuses that contain important neurovascular structures, which often preclude full surgical access for tumor resection. These PitNETs are difficult to treat and are referred to as aggressive PitNETs [27,28,29].

Treatment options remain limited for surgery-, chemotherapy- and radiation-refractory BTCNS, as well as malignant tumors. These tumors seriously affect the quality of life. Therefore, this article reviews the utility of DR for refractory BTCNS (meningiomas, schwannomas and PitNETs), which have higher incidence rates compared with other benign tumors of the CNS. All medicines can cause unwanted side effects. Possible side-effects are also reviewed for each candidate for DR.

The estimated time and steps of traditional de novo drug discovery exceed that of drug repositioning for the treatment of tumors. De novo drug discovery takes a long time to reach drug registration. In contrast, drug repositioning relatively takes a short time because it can bypass some processes if the antitumor potential of the drug is confirmed.

## 2. Meningioma

### 2.1. Antiseizure Drugs

In general, valproic acid is used for patients with epilepsy, bipolar disorder, and metabolic disease and prevention of migraine headaches [30]. Diarrhea, dizziness, drowsiness, hair loss, blurred/double vision, change in menstrual periods, ringing in the ears, shakiness (tremor), unsteadiness, and weight change may be reported as side effects [30]. Valproic acid reduced the expression of *Oct4* and anchorage-independent growth in meningioma. In the previous study, valproic acid inhibited the growth of cultured meningioma stem-like cells. Furthermore, valproic acid promoted radiosensitivity in those cells [31].

### 2.2. Anthelminthic Drugs

Mebendazole is a benzimidazole anthelmintic drug used to treat helminth infections [32]. The most common adverse effects accompanying mebendazole use are loss of appetite, abdominal pain, diarrhea, flatulence, nausea, vomiting, headache, tinnitus, and elevated liver enzymes. A small percentage of patients may experience convulsions, and some patients may show hypersensitivity reactions such as rash, urticaria, and angioedema [32]. Many studies showed that mebendazole inhibits tumor cell growth in various types of cancers [33]. After adding mebendazole, the hedgehog signaling pathway was inhibited and hedgehog-dependent survival was prevented in meningioma cell lines [34]. Mebendazole also blocked the VEGF receptor 2, leading to the inhibition of tumor angiogenesis in meningioma [35]. Furthermore, mebendazole caused mitotic arrest via depolymerization of tubulin and abnormal spindle cell formation [36]. A proapoptotic effect of mebendazole was also shown on meningioma cell lines [37].

### 2.3. Antidiabetic Drugs

Metformin is primarily used for the treatment of type 2 diabetes mellitus [38]. Nausea, vomiting, stomach upset, diarrhea, weakness, or a metallic taste in the mouth may occur as side effects [38]. Previous studies have demonstrated that metformin inhibits tumor stem cell growth in various types of cancers [39,40]. The effects of metformin are caused via the activation of AMP-activated protein kinase and the inhibition of the mitochondrial respiratory chain [40,41]. Metformin inhibited the mammalian target of the rapamycin signaling pathway [42]. Metformin also increased the therapeutic power of doxorubicin in tumor stem cells [43]. These actions of mechanisms may be applied for the treatment of meningiomas. Furthermore, metformin increased the efficacy of other anticancer drugs including cisplatin for meningioma [42,44]. The combination of metformin and chemotherapeutic drugs may improve treatment efficacy of patients with tumors including meningiomas.

### 2.4. Antihypertensive Drugs

Angiotensin II (ATII), which binds to the ATII type 1 receptor (ATIIR1) and ATIIR2, is the main effector of the renin–angiotensin system (RAS). ATIIR1 induces angiogenesis and cell proliferation [45]. Prorenin is also an important component of the RAS implicated in carcinogenesis through the Wnt/β-catenin signaling pathway [46]. The prorenin receptor, angiotensin converting enzyme, ATIIR1, and ATIIR2, which are components of the RAS, were expressed on the stem cell population in meningiomas, suggesting that these stem cells may be a potential therapeutic target through the inhibition of the RAS [45,47]. β-blockers, angiotensin-converting enzyme inhibitors, and angiotensin receptor blockers reduced tumor cell growth in various types of cancer, suggesting a potential utility of the repositioning of these drugs for the treatment of meningioma [48]. β-blockers are usually tolerated well without significant side effects. β-blockers may cause cold hands and feet. Some patients experience fatigue, perhaps related to excessive slowing of the heart rate. Erectile dysfunction may be a problem for males who take β-blockers. Healthcare providers often prescribe angiotensin receptor blockers instead of angiotensin-converting enzyme inhibitors because angiotensin receptor blockers cause fewer side effects. The main side effect of these drugs is dizziness. These drugs rarely affect blood flow to the kidneys. Chronic cough, loss of taste, and skin rash are also reported as side effects.

Calcium channel blockers (CCBs) are also a candidate for repositioning. Common side effects of CCBs include constipation, reduced heart rate, and fatigue. Dizziness, flushing, and edema are also possible side effects. However, side effects of CCBs are often mild and temporary. Cytosolic Ca^2+^ plays an important role in intracellular signal transduction, cell proliferation, and survival. Therefore, this ion can be a possible target in the treatment of cancer [49,50,51]. In meningioma, a dose-dependent decrease of tumor cells was shown when cultured with a CCB such as verapamil, nifedipine, or diltiazem [51,52]. In a similar study, diltiazem and verapamil decreased the growth of meningioma in vivo [53]. Verapamil or diltiazem added to hydroxyurea enhances in vitro and in vivo meningiomas growth inhibition [54,55]. However, this combination therapy failed to demonstrate significant radiographic response in a clinical trial [56].

### 2.5. Antihyperlipidemic Drugs

Hydroxymethylglutaryl-CoA (HMG-CoA) reductase inhibitors, also known as statins, are used to treat hypercholesterolemia. HMG-CoA reductase inhibitors block the conversion of HMG-CoA to mevalonate [57]. Commonly reported side effects of statins include muscle pain, headache, and stomach-related effects such as nausea, vomiting, and diarrhea. Statins causing serious side effects is extremely rare. Several studies have found that these drugs inhibit tumor cell proliferation [57,58]. For example, imvastatin showed downregulation of the PI3K/Akt pathway, leading to the inhibition of glioma cell proliferation [58]. A previous study has demonstrated the effects of lovastatin on meningioma cell proliferation and its influence on the MEK-1–MAPK/ERK pathway [57]. In another study, meningioma cell lines were treated with different combinations of statins and thiazolidinediones, which are medications used to manage and treat type 2 diabetes mellitus. The effect of simvastatin, lovastatin, atorvastatin, pravastatin, simvastatin, and two thiazolidinediones, pioglitazone and rosiglitazone, and their combinations were evaluated on human meningioma cell lines. Simvastatin was the most effective drug. Pioglitazone exhibited a significant effect. Furthermore, simvastatin in combination with pioglitazone was the most effective treatment for meningioma [59].

## 3. Schwannoma

### 3.1. Anti-Inflammatory Drugs

The expression of cyclooxygenase-2 (COX-2) is related to the proliferation of sporadic and NF2-related vestibular schwannoma [60]. NF2 gene mutation activates the Hippo pathway. The effector molecule YAP promotes the transcription of the COX-2 for the prostaglandin biosynthesis. Prostaglandin E2 catalyzed by COX-2 demonstrates some roles in both cell proliferation and angiogenesis [61]. Therefore, COX-2 inhibitors have the possibility to inhibit the growth of vestibular schwannoma. A negative correlation between aspirin users and the growth of sporadic vestibular schwannoma was shown [62]. However, other studies on celecoxib, aspirin, and COX-2 inhibitors in non-steroidal anti-inflammatory drugs (NSAIDs) demonstrated that no growth inhibitory effect was shown for celecoxib on NF2 or aspirin on the patients with vestibular schwannoma [63,64,65]. Other studies showed that NSAIDs, glucocorticoids, or other immunosuppressive drugs could not alter the COX-2 expression in vestibular schwannoma. Several studies demonstrated that NSAIDs were not associated with the growth of vestibular schwannoma [65,66,67]. Aspirin can also suppress the NF-κB pathway in vestibular schwannoma, leading to another mechanism of treatment [68,69]. Although there is still debate on the matter, aspirin may be used for a “wait and scan” approach for patients with vestibular schwannoma [70]. NSAIDs have well-known adverse effects affecting the gastric mucosa, renal system, cardiovascular system, hepatic system, and hematologic system.

Hearing disturbance determines quality of life in patients with vestibular schwannoma. Among the list of dysregulated genes in vestibular schwannoma, neuroinflammation-related signaling was one of the highest ranked pathways [71]. The activation of the NLR family pyrin domain containing 3 (NLRP3) inflammasome, a multi-protein complex that activates caspase-1 resulting in the production of inflammatory cytokines, has been related to the inner ear biology. The NLRP3 mutation is associated with cochlear autoinflammation in conjunction with deafness, autosomal dominant 34, with or without inflammation (DFNA34)-mediated hearing loss and age-rated hearing loss. Activation of NLRP3 triggers the production of interleukin (IL)-1β, which is a potent proinflammatory cytokine. A recombinant human IL-1 receptor antagonist reversed the hearing loss observed in a family with sensorineural hearing loss and NLRP3 mutations [72,73]. Anakinra, an IL-1 receptor antagonist, has been previously used in the treatment of rheumatoid arthritis and autoinflammatory syndromes. This drug may be a DR drug for the treatment of vestibular schwannoma. Commonly reported side effects of anakinra include infection, antibody development, and inflammation at the injection site.

### 3.2. Abortion Pills

Mifepristone (RU486) is a progesterone and glucocorticoid receptor antagonist used for medical abortion [74]. Side effects of mifepristone may include dizziness, weakness, vomiting, headache, diarrhea, nausea, and fever or chills. The computational drug repositioning platform found that mifepristone has the potential to treat vestibular schwannomas [71]. Mifepristone acts on the upstream of vestibular schwannoma inflammation markers, such as NF-κB [75]. This effect was independent of whether the NF2 gene was mutated [75]. A phase II clinical trial on mifepristone in vestibular schwannoma is currently being planned.

## 4. Pituitary Neuroendocrine Tumor

### 4.1. Antidiabetic Drugs

In general, tumor stem cells are derived from normal tissue specific cells or re-differentiate and/or de-differentiate from progenitor/differentiated cells. Some studies have discussed the relationship between genetic alterations in human pituitary gland stem cells and tumors formed from pituitary adenoma stem cells (PASCs) [76,77]. While stem-like cells are particularly resistant to classical cytotoxic drugs due to the overexpression of drug-extruding pumps and DNA-repairing enzymes [77], many previous studies demonstrated a higher sensitivity of tumor stem cell subpopulation to metformin [78], which may lead to the development of a DR approach in PitNETs [79,80]. Because growth hormone-secreting PitNET (GH-PitNET) patients have a high incidence of diabetes, these patients are frequently treated with metformin. In the previous study, metformin significantly inhibited GH-PitNET cell proliferation and GH secretion in vivo [79]. These findings suggest metformin may be a promising therapeutic agent for the treatment of GH-PitNET, particularly in patients with diabetes. The side effects of metformin are described above.

### 4.2. Antabuse

GH-PitNET is characterized by the increased volume of limbs, lower jaw protrusion, macroglossia, and increase of GH and insulin-like growth factor-1 levels [81]. PASCs showed upregulation of the multidrug transporters, such as ABC transporter subfamily B member 1 and ATP-binding cassette transporter G2 [82], suggesting their ability to extrude cytotoxic drugs. PASCs from a GH-PitNET were insensitive to both carboplatin, etoposide, and temozolomide [83,84]. In vitro and in vivo treatment with disulfiram, a clinically approved drug for the treatment of alcoholism, sensitizes PASCs to temozolomide cytotoxicity, preventing drug-induced DNA damage repair by inhibiting O-6-methylguanine-DNA methyltransferase expression [85]. Common side effects of disulfiram include skin rash, acne, headache, tiredness, impotence, and a metallic or garlic-like taste in the mouth.

### 4.3. Nonsteroidal Anti-Estrogenic Agent

Tamoxifen (a nonsteroidal anti-estrogenic agent) is used mainly to treat hormone receptor-positive breast cancer (breast cancer with cells that have estrogen and/or progesterone receptors on them) [86]. Common tamoxifen-attributed side effects are hot flashes, vaginal dryness, sleep problems, weight gain, depression, irritability, and mood swings. The previous study repositioned the immune regulator tamoxifen to target the signal transducer and activator of transcription 6 (STAT6) in PitNETs. Tamoxifen inhibited the proliferation of PASCs. Tamoxifen downregulated phosphorylated PI3K, phosphorylated AKT, and the anti-apoptotic protein Bcl-2 and increased the expression of pro-apoptotic proteins p53 and Bax in PitNET cells. Furthermore, tamoxifen also inhibited the migration of both cell lines by reprogramming tumor-associated macrophages to the M1 phenotype through STAT6 inactivation and inhibition of the macrophage-specific immune checkpoint molecule [87]. Therefore, tamoxifen is likely to be a promising combination therapy for PitNETs [88].

### 4.4. Other Drugs (Combination Therapy)

Cushing’s disease (CD) is a disease caused by adrenocorticotropic hormone (ACTH) over-secretion from the adrenocorticotropic adenomas, leading to stimulation of the adrenal gland to secrete cortisol [89]. However, ACTH-producing PitNET-directed drugs can only inhibit ACTH secretion, limiting clinical applications. Recently, Nur77 (also known as NR4A1 and NGFI-Ba, a kind of nuclear receptor), a positive transcription regulator of pro-opiomelanocortin (POMC), the precursor of ACTH, has been regarded as a target of CD [90]. The previous study repositioned FDA-approved bexarotene (BEXA), which is a drug for the cutaneous T-cell lymphoma. BEXA can be used for the treatment of CD based on artificial intelligence prediction. The combination of BEXA and the FDA-approved targeted epidermal growth factor receptor drug (lapatinib, LAPA) was selected based on the analysis of computational cell signaling transduction. The synergistic inhibition of BEXA/LAPA on ACTH-producing PitNET growth was made by promoting the formation of the Nur77- retinoid X receptor-α dimer, leading to hormone normalization and tumor growth suppression [91]. Therefore, a novel combination therapy of BEXA/LAPA through a Nurr77-dependent mechanism was suggested as a treatment for CD. Hyperlipidemia, headache, weakness, leukopenia, anemia, infection, dry skin, rash, and photosensitivity are known as common side effects of BEXA. Cataracts, neutropenia, hypothyroidism, pancreatitis, and hepatitis are known to be rare but severe adverse events. Nausea, vomiting, upset stomach, mouth sores, mild rash, dry skin, and trouble sleeping may occur as side effects of LAPA.

Drug repositioning can be achieved through literature search, screening collections of drugs in vitro, or computers and artificial intelligence (AI). Drug repositioning was frequently used for the malignant brain tumors including metastatic brain tumor, malignant glioma, and central nervous system malignant lymphoma. This strategy can be applied for refractory benign brain tumors including Grade 2 meningiomas, NF2-related schwannomas, and PitNET with cavernous sinus invasion.

## 5. Future Directions

Surgery is the cornerstone of treatment for BTCNS. Some BTCNS display aggressive behavior characterized by multiple recurrences and invasion of the brain, dura, and adjacent tissue. This aggressive or malignant phenotype of BTCNS is difficult to treat, similar to malignant tumors. There are no treatment options for patients with inoperable tumors or those who are medically unsuitable for surgery. In general, conventional radiation therapy and/or chemotherapeutic agents are not beneficial for those with aggressive and refractory behavior.

Although the side effects of DR are less severe than those of chemotherapy and molecular targeted therapy, DR has had limited effects on malignant tumor. Stable disease may be one of the important goals for a DR strategy. BTCNS tend to grow slowly. In the setting of refractory BTCNS, prolonged stable disease has the possibility to represent a meaningful benefit, which is associated with improvement in symptoms and enhanced quality of life. Furthermore, treatment of rare and intractable diseases, minimizing attrition rates, reducing the cost of therapy, etc. are the advantages of DR. This article emphasizes the clinical potential and application of a DR strategy for refractory BTCNS (Figure 2).

Because few studies have previously reported in the area of a DR approach for benign tumors, the most specific drug option is unclear for refractory BTCNS. In particular, antiseizure drugs may be a good option for DR for refractory BTCNS. Although the risk of a seizure varies with the tumor type and its location, up to 60% of people with brain tumors present with seizures or may have a seizure after surgery [92]. Seizures are an added burden with a negative impact on the quality of life. Therefore, a DR strategy using antiseizure drugs has a possibility of controlling not only seizures but also tumor growth, leading to improvement of the quality of life.

There are almost no strict clinical trials using a DR strategy against refractory BTCNS. Therefore, we must confirm the dosage regimen of those tumors using preclinical and clinical trials [93,94]. The dosage of those drugs may not be the same as was used during its primary treatment. In general, the side effects of DR are less severe than those of chemotherapy and molecular targeted therapy. We should carefully consider the possibility of side effects for patients with BTCNS who do not have the primary disease against the DR drugs.

Basic research is essential to find out what treatments work best for what patient. A DR strategy can accelerate clinical trials, which are an essential part of the research and development of a new treatment strategy, bringing benefits to patients with refractory BTCNS.

## Figures and Tables

**Figure 1 ijms-24-12997-f001:**
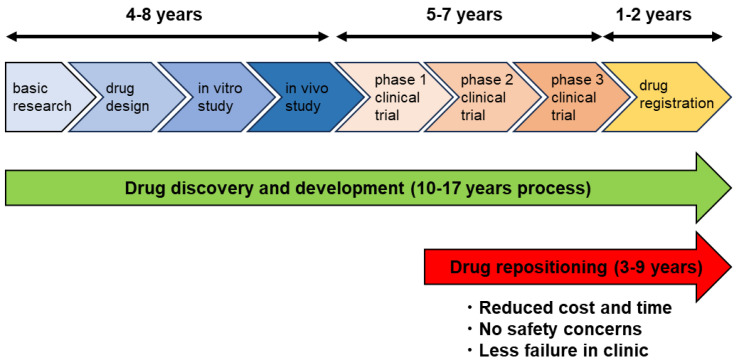
Drug discovery process through drug repositioning.

**Figure 2 ijms-24-12997-f002:**
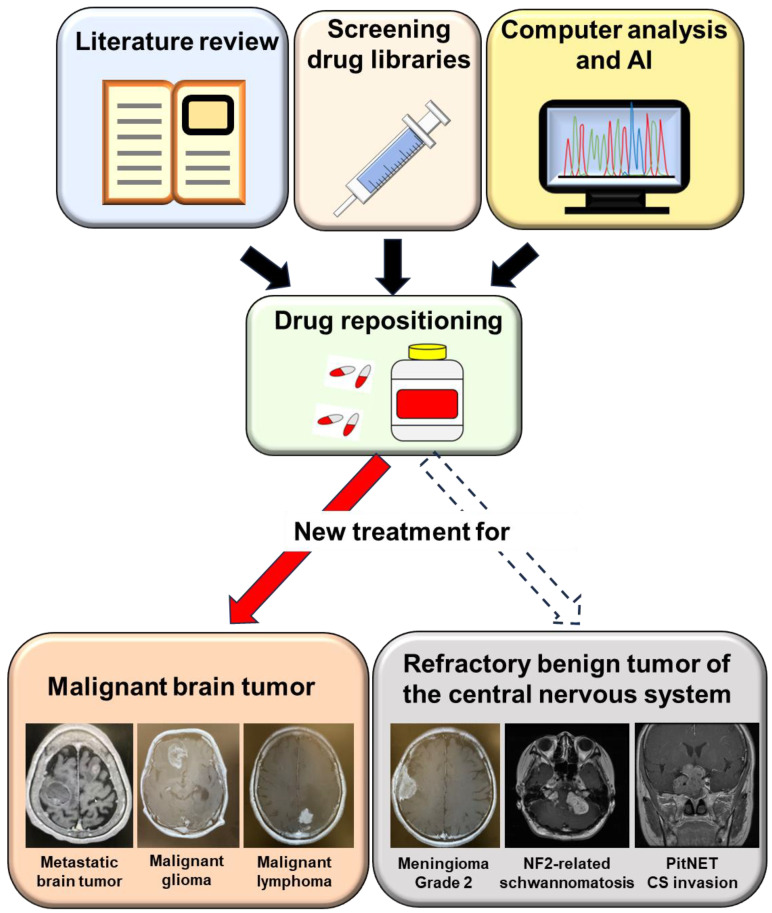
Drug repositioning for the refractory benign tumors of the central nervous system.

## Data Availability

Not applicable.

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
