# Peer review of "Drug Repositioning for Refractory Benign Tumors of the Central Nervous System"

_ijms, 2023, doi:10.3390/ijms241612997_

Round 1
Reviewer 1 Report
In the review by Ryota Tamura, a review of drug repositioning (DR) is presented in regards to select benign brain tumors of the central nervous system (CNS). A variety of drugs with approved clinical use are presented as possible therapeutic options for benign CNS tumors that are refractory to surgery and other conventional treatment approaches. In particular, is reviewed in respect to meningiomas, schwannomas, and pituitary neuroendocrine tumors (PitNETs). While the concept is interesting, there are several aspects of this manuscript which can be improved.
- The authors chose relatively common benign CNS tumors, however, several other important benign brain tumors are not mentioned, including pilocytic astrocytoma and subependymoma, to name just a couple. It should be clarified why certain tumors were chosen for review and not others.
- The term "benign brain tumors (BBTs)" is used throughout the manuscript. However, this reviewer considers "brain tumors" to include entities that arise in the brain itself rather than its surrounding structures. Meningioma is a dural neoplasm, schwannoma is a nerve sheath tumor, and PitNET is a neuroendocrine tumor of the adenohypophysis, none of which are truly "brain" tumors. The title is more correct in describing them as benign tumors of the central nervous system. This terminology should be used unless additional brain tumors (benign astrocytomas, etc., as discussed above) are included.
- Figure 1 illustrates the relative process of drug discovery and development but there is no indication of the actual amount of time the various components take. Objective data should be included in the text of the average/median time frame (or case examples if averages are not available) of drug development both de novo and using DR. Similarly, differences in cost between de novo and DR should be discussed.
- In the abstract, Grade 3 meningioma is described incorrectly as a benign brain tumor. However, Grade 3 (anaplastic) meningioma is a malignant CNS tumor.
- Finally, while the English language of the manuscript is understandable, there are many instances of awkward word use, strange transitional phrases, and incorrect use of articles and punctuation. Professional English language editing services should be considered if English is not the author's primary language.
See above.
Author Response
We are very grateful to the reviewers for their insightful comments and suggestions, which would undoubtedly help us to improve our manuscript immensely. As indicated in the responses below, we have taken all their comments and suggestions into account when generating the revised version of the manuscript. Responses to the reviewers’ comments appear after the arrows, in blue text.
Reviewer 1:
1.In the review by Ryota Tamura, a review of drug repositioning (DR) is presented in regards to select benign brain tumors of the central nervous system (CNS). A variety of drugs with approved clinical use are presented as possible therapeutic options for benign CNS tumors that are refractory to surgery and other conventional treatment approaches. In particular, is reviewed in respect to meningiomas, schwannomas, and pituitary neuroendocrine tumors (PitNETs). While the concept is interesting, there are several aspects of this manuscript which can be improved.
→
Thank you very much for your review.
2.The authors chose relatively common benign CNS tumors, however, several other important benign brain tumors are not mentioned, including pilocytic astrocytoma and subependymoma, to name just a couple. It should be clarified why certain tumors were chosen for review and not others.
→
Thank you very much for your comments.
Meningiomas, PitNETs, and schwannomas are the most common benign tumors of the central nervous system.
The most common tumors were meningiomas (46.3%), followed by gliomas (17.2%), PitNETs (13.1%), schwannomas (8.2%), and malignant lymphomas (3.8%) [Miller KD.CA Cancer J Clin. 2021;Matsumoto F. Neurol Med Chir (Tokyo). 2021]. However, treatment options for surgical, chemotherapy and radiation-refractory benign tumor of the central nervous system remain limited as well as malignant tumors. Therefore, we have reviewed the DR strategy for meningiomas, schwannomas and PitNETs which have higher incidence rates compared with other benign tumors of the central nervous system.
Miller KD, Ostrom QT, Kruchko C, Patil N, Tihan T, Cioffi G, Fuchs HE, Waite KA, Jemal A, Siegel RL, Barnholtz-Sloan JS. Brain and other central nervous system tumor statistics, 2021.CA Cancer J Clin. 2021;71:381-406.
Matsumoto F, Takeshima H, Yamashita S, Yokogami K, Watanabe T, Ohta H; Miyazaki Brain Tumor Research Group. Epidemiologic Study of Primary Brain Tumors in Miyazaki Prefecture: A Regional 10-year Survey in Southern Japan. Neurol Med Chir (Tokyo). 2021;61:492-498.
3.The term "benign brain tumors (BBTs)" is used throughout the manuscript. However, this reviewer considers "brain tumors" to include entities that arise in the brain itself rather than its surrounding structures. Meningioma is a dural neoplasm, schwannoma is a nerve sheath tumor, and PitNET is a neuroendocrine tumor of the adenohypophysis, none of which are truly "brain" tumors. The title is more correct in describing them as benign tumors of the central nervous system. This terminology should be used unless additional brain tumors (benign astrocytomas, etc., as discussed above) are included.
→
Thank you very much for your comments. According to the reviewer’s comments, we have used the “benign tumors of the central nervous system (BTCNS)”.
- Figure 1 illustrates the relative process of drug discovery and development but there is no indication of the actual amount of time the various components take. Objective data should be included in the text of the average/median time frame (or case examples if averages are not available) of drug development both de novo and using DR. Similarly, differences in cost between de novo and DR should be discussed.
→
Thank you very much for your comments. We have added the average time frame of de novo and DR strategy. Differences in cost between de novo and DR was discussed in the revised manuscript.
The discovery process for de novo drugs is costly and time-consuming.
The average cost for a de novo discovery and development of a drug is approximately US$2.5 billion and requires around 10-17 years to reach the market [Paul SM. Nat Rev Drug Discov. 2010; Hay M. Nat Biotechnol. 2014; Nosengo N. Nature. 2016]. In case of DR process, it costs around ≤60% expenditure of a de novo discovery [Napolitano F. J Cheminform. 2013].
Paul SM, Mytelka DS, Dunwiddie CT, et al. How to improve R&D productivity: the pharmaceutical industry’s grand challenge. Nat Rev Drug Discov. 2010; 9: 203-214.
Hay M, Thomas DW, Craighead JL, Economides C, Rosenthal J. Clinical development success rates for investigational drugs. Nat Biotechnol. 2014; 32: 40-51.
Nosengo N. Can you teach old drugs new tricks? Nature. 2016; 534: 314-316.
Napolitano F, Zhao Y, Moreira VM, Tagliaferri R, Kere J, D'Amato M, Greco D. Drug repositioning: a machine-learning approach through data integration. J Cheminform. 2013 Jun 22;5(1):30.
5.In the abstract, Grade 3 meningioma is described incorrectly as a benign brain tumor. However, Grade 3 (anaplastic) meningioma is a malignant CNS tumor.
→
Thank you very much for your comments. We have deleted Grade 3 meningioma in the abstract.
6.Finally, while the English language of the manuscript is understandable, there are many instances of awkward word use, strange transitional phrases, and incorrect use of articles and punctuation. Professional English language editing services should be considered if English is not the author's primary language.
→
Thank you very much for your comments. The native speaker has already edited a draft of this manuscript. In addition, revised manuscript was also edited by the native speaker. We have modified the manuscript again.
Reviewer 2 Report
Major Comments:
1. The authors stated about many treatment options as DR for each of the three BBTs. Then, what specific drug option (DR) can be the ideal for each of those three BBTs? The authors should describe about this concern.
2. What will be the dosage of those drugs (DR) in the treatment of BBTs? Will it be the same as it is used during its primary treatment?
3. The authors should discuss whether these drugs (DR) can be used against BBTs even the patient doesn’t have the disease against which those drugs are primarily used? In this case, will there be any side effects?
4. Are there any reported clinical trials of the DR against BBTs?
5. As the cancer research is so dynamic and many treatment options are emerging continuously, do the authors still think that DR could be the better option compared to those specific anticancer drugs? Please discuss.
Major Comments:
1. The authors stated about many treatment options as DR for each of the three BBTs. Then, what specific drug option (DR) can be the ideal for each of those three BBTs? The authors should describe about this concern.
2. What will be the dosage of those drugs (DR) in the treatment of BBTs? Will it be the same as it is used during its primary treatment?
3. The authors should discuss whether these drugs (DR) can be used against BBTs even the patient doesn’t have the disease against which those drugs are primarily used? In this case, will there be any side effects?
4. Are there any reported clinical trials of the DR against BBTs?
5. As the cancer research is so dynamic and many treatment options are emerging continuously, do the authors still think that DR could be the better option compared to those specific anticancer drugs? Please discuss.
Author Response
We are very grateful to the reviewers for their insightful comments and suggestions, which would undoubtedly help us to improve our manuscript immensely. As indicated in the responses below, we have taken all their comments and suggestions into account when generating the revised version of the manuscript. Responses to the reviewers’ comments appear after the arrows, in blue text.
Reviewer 2:
- The authors stated about many treatment options as DR for each of the three BBTs. Then, what specific drug option (DR) can be the ideal for each of those three BBTs? The authors should describe about this concern.
→
Thank you very much for your review.
Because few studies have previously reported for benign tumor, the most specific drug option is unclear. In our opinion, antiseizure drugs may be a good option of DR for refractory benign tumors of the central nervous system. Although the risk of a seizure varies with the tumor type and its location, up to 60% of people with brain tumors present with seizures, or may have a seizure after surgery [Tremont-Lukats IW. Cofhrane Database Syst Rev. 2008]. Seizures are an added burden with a negative impact on quality of life. Therefore, DR strategy using antiseizure drugs has a possibility to control not only seizures but also tumor growth, leading to improvement of quality of life.
Tremont-Lukats IW, Ratilal BO, Armstrong T, Gilbert MR. Antiepileptic drugs for preventing seizures in people with brain tumors. Cochrane Database Syst Rev. 2008;2008:CD004424.
- What will be the dosage of those drugs (DR) in the treatment of BBTs? Will it be the same as it is used during its primary treatment?
→
Thank you very much for your comments. We have added the following discussion in the revised manuscript.
Because a lot of studies of DR have been reported for malignant tumor, there are no clinical trials using DR strategy against refractory benign tumor of the central nervous system. Therefore, we must confirm the dosage regimen of those tumors using the strict preclinical and clinical trials [Opera TI Assay Drug Dev Technol. 2015; Oprea TI. The AAPS Journal. 2012]. Probably, the dosage of those drugs is the same as it is used during its primary treatment.
Oprea TI, Overington JP. Computational and practical aspects of drug repositioning. Assay Drug Dev Technol. 2015;13:299-306.
Oprea TI, Mestres J. Drug Repurposing: Far Beyond New Targets for Old Drugs. The AAPS Journal. 2012;14: 759-763.
- The authors should discuss whether these drugs (DR) can be used against BBTs even the patient doesn’t have the disease against which those drugs are primarily used? In this case, will there be any side effects?
→
Thank you very much for your comments.
We have added the statement for the patients who do not have the primary disease against the DR drugs. In addition, we have described the possible side effects for each drug in the manuscript.
In general, the side effects of DR are less severe than those of chemotherapy and molecular targeted therapy. We should carefully consider the possibility of side effects for the patients who do not have the primary disease against the DR drugs.
- Are there any reported clinical trials of the DR against BBTs?
→
Although a lot of studies of DR have been reported for malignant tumor, there are no clinical trials using DR strategy against refractory benign tumor of the central nervous system.
We have added the information in the revised manuscript.
- As the cancer research is so dynamic and many treatment options are emerging continuously, do the authors still think that DR could be the better option compared to those specific anticancer drugs? Please discuss.
→
Thank you very much for your review. We have added the following discussion in the revised manuscript.
Although the side effects of DR are less severe than those of chemotherapy and molecular targeted therapy, the DR had limited effect for malignant tumor.
Stable disease may be one of the important goals for DR strategy. Benign tumors in the central nervous system tend to grow slowly. In the setting of refractory benign tumors of the central nervous system, stable disease may provide significant benefit to the patient. Prolonged stable disease has the possibility to represent a meaningful benefit, which is associated with improvement in symptoms and enhanced quality of life. Therefore, DR is thought to be appropriate for refractory benign tumor of the central nervous system.